# A biomimetic electrostatic assistance for guiding and promoting N-terminal protein chemical modification

Nathalie Ollivier[1], Magalie Sénéchal[1], Rémi Desmet [1], Benoît Snella[1], Vangelis Agouridas [1,2] & Oleg Melnyk [1]✉

The modification of protein electrostatics by phosphorylation is a mechanism used by cells to promote the association of proteins with other biomolecules. In this work, we show that introducing negatively charged phosphoserines in a reactant is a powerful means for directing and accelerating the chemical modification of proteins equipped with oppositely charged arginines. While the extra charged amino acid residues induce no detectable affinity between the reactants, they bring site-selectivity to a reaction that is otherwise devoid of such a property. They also enable rate accelerations of four orders of magnitude in some cases, thereby permitting chemical processes to proceed at the protein level in the low micromolar range, using reactions that are normally too slow to be useful in such dilute conditions.

A large body of evidence coming from computational and experimental studies shows the critical role played by electrostatics in the binding mechanism of proteins to a variety of biomolecules[1]. For example, long-range attractive electrostatic forces can increase the rate of protein-protein association by many orders of magnitude[2]. Engineering the surface charge of proteins was successfully used for improving the binding strength of protein complexes[3] or for promoting or modifying the biochemical activity of enzymes[4,5]. The mimicry of such mechanisms has led to the development of electrostatic assistance in synthetic organic chemistry[6], which consists in equipping the reactants with ions of opposite charge to assist their reaction and the formation of a covalent bond between them. Electrostatic assistance proved efficient for promoting the reaction of small reagents with polymers presenting a large number of the same complementary functional entity[6,7]. Unfortunately, its use for assisting the selective modification of protein molecules in water as depicted in Fig. 1a has shown only modest effects so far, even with the modifier in excess and the target protein in the high micromolar concentration range[8]. This is unfortunate because using attractive electrostatic forces for assisting chemical reactions between protein molecules can potentially benefit from several advantages. The first is that some charged amino acids can be programmed into proteins using standard genetic methods. Long-range electrostatic forces can also be engineered with a limited number of amino acid residues, as opposed to the proximity-directed approaches that depend on large biomolecular templates and the establishment of a tight binding interface for bringing reactants close in space[9]. Although electrostatic assistance could have a broad scope, its use for selective protein modification raises the question whether the method can work with protein domains, which are intrinsically decorated with a plethora of charged amino acid residues.

In search of a solution to implement electrostatic assistance with proteins, we were inspired by the capacity of cells to promote or inhibit the association of proteins with other biomolecules through multiple phosphorylation of poorly structured domains[10]. Each phosphorylation event adds two negative charges to the protein, and the global effect depends more on the number of phosphate groups than on their position. Therefore, we primarily used phosphoserine (pSer) residues to provide negative charges to the reactants, typically in the form of a tripeptidic pSer–Gly–pSer module (Fig. 1b). Because the programming of pSer in live cells is a difficult task[11,12], we also evaluated the interest of glutamic acid residue as the negative charge carrier in some studies. On the other side, the design of the positively charged module was suggested by the capacity of arginine-rich peptides to interact through their cationic side chain guanidinium groups with a variety of negatively charged biomolecules, including phospholipids.

[1]Univ. Lille, CNRS, Inserm, CHU Lille, Institut Pasteur de Lille, U1019 - UMR 9017; Center for Infection and Immunity of Lille, F-59000 Lille, France. [2]Centrale Lille, F-59000 Lille, France. ✉e-mail: oleg.melnyk@ibl.cnrs.fr

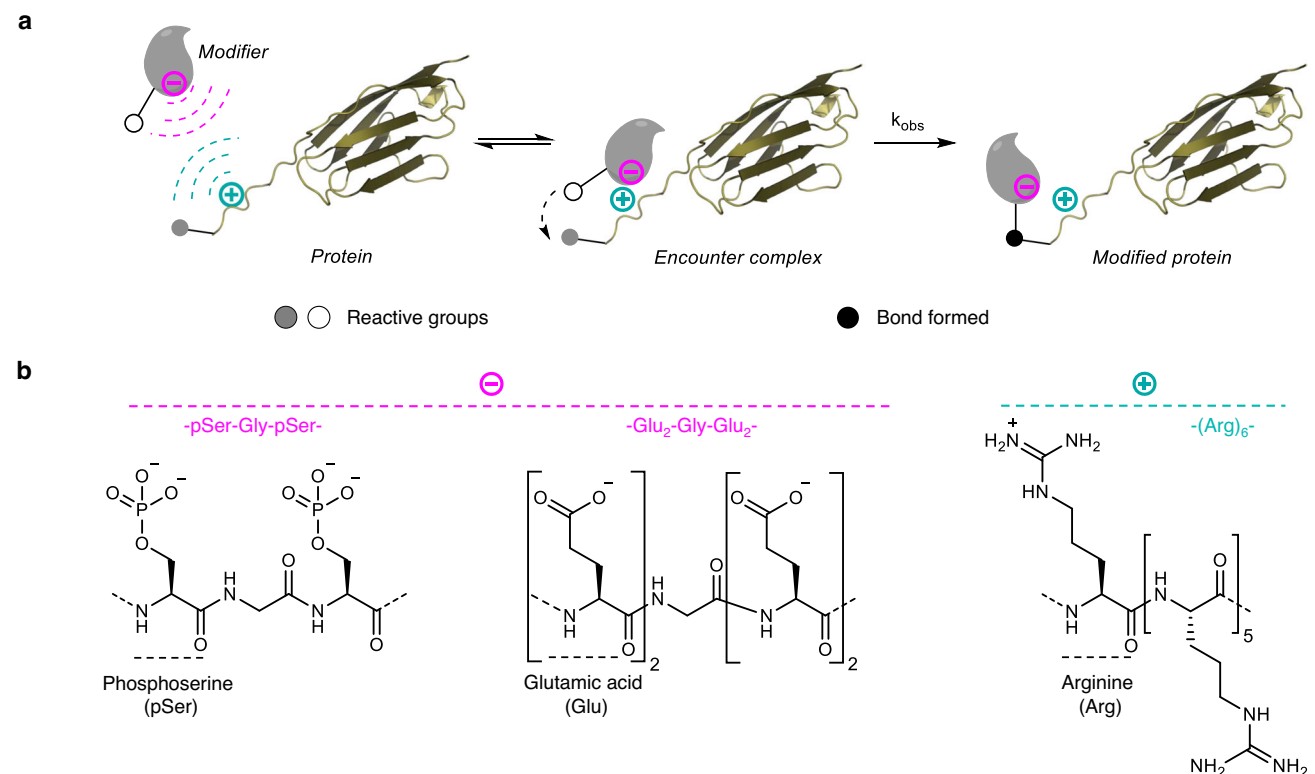

**Fig. 1 | Electrostatic assistance of covalent protein modification. a** Principle of electrostatic assistance. **b** Phosphoserine (pSer) or glutamic acid (Glu) residues were used in this work to provide negative charges to the reactants, while the positive charges were brought by arginine (Arg) residues.

Therefore, we used a cationic module made of contiguous arginines (Fig. 1b). To implement the electrostatic assistance concept, we used the peptide thioester aminolysis reaction, a chemical process that is notoriously poorly selective and inefficient in water in the mM concentration range. As such, it is nicely adapted for judging the potential of the method to promote complex chemical transformations at the protein level. We next used the electrostatic assistance to promote reactions enabling the selective N-terminal chemical modification of proteins, a modification site that is often exploited for protein conjugate synthesis[13–16]. The first studied reaction of this kind is the native chemical ligation between a C-terminal peptide thioester and a N-terminal cysteinyl peptide (NCL[17]). This reaction is intensively used nowadays for the chemical synthesis or site-specific modification of proteins[18–20]. The second ligation method that was subjected to electrostatic assistance is hydrazone ligation. The ease of accessing protein hydrazide and aldehyde reactants makes hydrazone ligation a popular conjugation method[21]. Indeed, protein hydrazides can be easily accessed by chemical[22], biochemical[23] or biological[24] means. The α-oxo aldehyde functionality can be easily installed at the N-terminus of proteins as well, typically by oxidation of a seryl residue or by transamination[25]. The importance of this chemistry for conjugate synthesis has stimulated the development of a variety of amine catalysts to promote hydrazone ligation at neutral pH[21], following the pioneering studies on nucleophilic catalysis of Schiff base formation by aniline[26,27].

We show here that introducing negative charges in a reactant in the form of phosphoserine or glutamic acid residues is a powerful means for directing and accelerating the N-terminal chemical modification of proteins equipped with oppositely charged arginines. The assistance is purely electrostatic in nature and induces no detectable affinity between the reactants. Electrostatic assistance brings site-selectivity to the peptide thioester aminolysis reaction and renders it synthetically useful on the protein level at sub-millimolar concentrations. Electrostatic assistance has also a marked effect on the NCL reaction with rate accelerations of four orders of magnitude. Such a rate acceleration permits the NCL reaction to proceed at the protein level in the low micromolar range, while the non-assisted reaction is too slow to be useful in such dilute conditions. Application of the electrostatic assistance to the non-native hydrazone ligation results as well in the accelerated formation of conjugates. We show that electrostatic assistance synergizes with amine catalysts to promote hydrazone formation. Taken together, our results suggest that electrostatic assistance can potentially promote a variety of ligation methods.

## Results

### Proof-of-concept for electrostatic assistance at the protein level

We started exploring the potential of phosphoserine-arginine electrostatic assistance by promoting the formation of peptide bonds by aminolysis of peptide thioesters in neutral water (Fig. 2). Note that the peptide thioester aminolysis reaction, which involves the reaction of a C-terminal peptide thioester with the α-amino group of another peptide segment, has been successfully used by Hojo and Aimoto for chemical protein synthesis[28]. This method uses silver ion in an organic solvent as a mean for activating the thioester and requires the protection of the amine side-chain functionality of lysine residues which, if left unprotected, would compete for the α-amino group[29].

The pSer module was inserted near the C-terminus of the peptide thioester, while the Arg module was inserted near the N-terminus of the peptide nucleophile (Fig. 2a). Regarding the buffer used for performing the reaction and although 50 mM sodium phosphate is well tolerated, we preferred to use sodium hydrogen carbonate/carbon dioxide as the buffering system because sodium phosphate buffer, which is more classically used for conducting peptide ligation reactions, competes with the phosphoserine residues (see Supplementary Methods and Supplementary Figs. 174 and 175). Moreover, phosphate

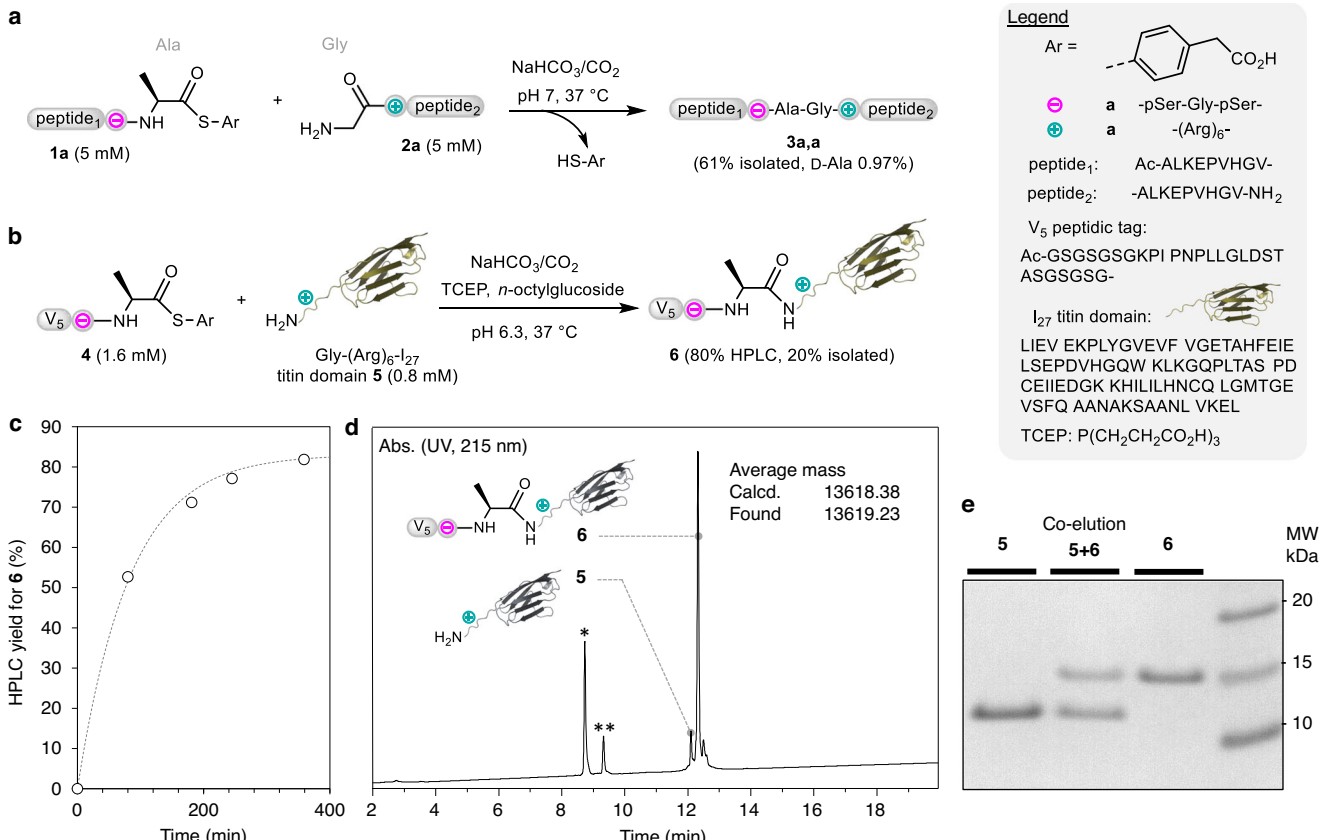

**Fig. 2 | Electrostatic assistance of peptide thioester aminolysis. a** The reaction of peptide thioester **1a** (5 mM) with glycyl peptide **2a** (5 mM) shows the formation of target peptide **3a,a** as the major product. The reaction was run in sodium hydrogen carbonate/carbon dioxide pH 7 buffer at 37 °C, which was obtained by equilibrating sodium hydrogen carbonate (50 mM) in a cell incubator (5% partial $CO_2$ pressure). **b** The reaction of peptide thioester **4** (1.6 mM) with Gly-$(Arg)_6$-$I_{27}$ titin protein **5** (0.8 mM) yielded target protein **6** as the major product. Conditions: Sodium bicarbonate/$CO_2$ buffer (20 mM), 37 °C, TCEP • HCl 1 mM, *n*-octylglucoside 10 mM, 20 h. **c** Kinetic monitoring of the formation of target protein **6**. **d** LC-MS of the crude mixture corresponding to the reaction shown in **b**. *peptide thioester hydrolysis byproduct, **peptide thioester cyclization byproduct (see Supplementary Methods for details). **e** SDS-PAGE analysis of purified target conjugate **6** (Coomassie blue staining).

buffers can promote the epimerization and hydrolysis of peptide aryl thioesters of the type used in this work[18]. Strikingly, the reaction of peptide aryl thioester **1a** derived from 4-mercaptophenylacetic acid[30] with one equivalent of peptide nucleophile **2a** at neutral pH in sodium hydrogen carbonate/carbon dioxide buffer resulted in the formation of product **3a,a** having an Ala-Gly peptidic bond with high chemoselectivity and in high yield (74% by HPLC, 61% isolated) (Fig. 2a). The byproducts generated by the reactivity of the side-chain amino group of lysine residues present in both peptide segments or from the epimerization of the peptide thioester were formed in insignificant amounts (see Supplementary Methods and Supplementary Figs. 59–63 and 186). Remarkably, decreasing peptide reactant concentration down to 0.1 mM still furnished peptide **3a,a** as the major product (60% by HPLC, see Supplementary Figs. 66 and 67). Because the presence of phosphorylated amino acid residues in the final product is not always desirable, we verified that the treatment of peptide **3a,a** with alcaline phosphatase resulted in its rapid dephosphorylation (see Supplementary Figs. 202 and 203).

To further explore the potential of phosphoserine-arginine electrostatic assistance at the level of a folded protein domain, the $I_{27}$ immunoglobulin-like titin domain equipped with the Gly-$(Arg)_6$-N-terminal extension was expressed in *E. coli*. The protein was reacted at 0.8 mM with only two equivalents of the peptide thioester **4**, in which a V5 peptidic tag is associated with the pSer module (Fig. 2b). The reaction proceeded remarkably well in forming the target conjugate **6** according to an apparent first-order rate law and with 80%

conversion (Fig. 2c–e). Notably, the reaction was found to be highly selective for the N-terminal amine (see Supplementary Figs. 204–210).

## Insight into the mechanism of electrostatic assistance

Having established the potential of phosphoserine-arginine electrostatic assistance at the level of a folded protein domain, we next used the peptide thioester aminolysis reaction on model peptides in order to determine the intricate interplay of reaction parameters, beginning with the role of module composition on electrostatic assistance (Fig. 3). The replacement of phosphoserine residues in the thioester by serines (**1d** + **2a**), the deletion of the $Arg_6$ module (**1a** + **2g**) or both (**1d** + **2g**) resulted in a dramatic reduction of the yields and in poor selectivities (Fig. 3a, b), unambiguously showing the importance of equipping both reaction partners with complementary charged modules. We also ran two different competition experiments by reacting an equimolar mixture of peptide thioesters **1a** and **1d** with peptide **2a** (Supplementary Methods, Supplementary Figs. 193 and 194), or peptide thioester **1a** with an equimolar mixture of peptides **2a** and **2g** (Supplementary Methods and Supplementary Figs. 195 and 196). These experiments furnished peptide **3a,a** as the only ligation product, showing that in the absence of the appropriate module a reactant is unable to compete with the assisted process.

Next, the impact of the number of arginines in the positively charged module was assessed by reacting peptide thioester **1a** with glycyl peptides **2a-g** (Fig. 3a, c). The yield of aminolysis product **3** was about the same when the cationic module contained 2–6 arginines

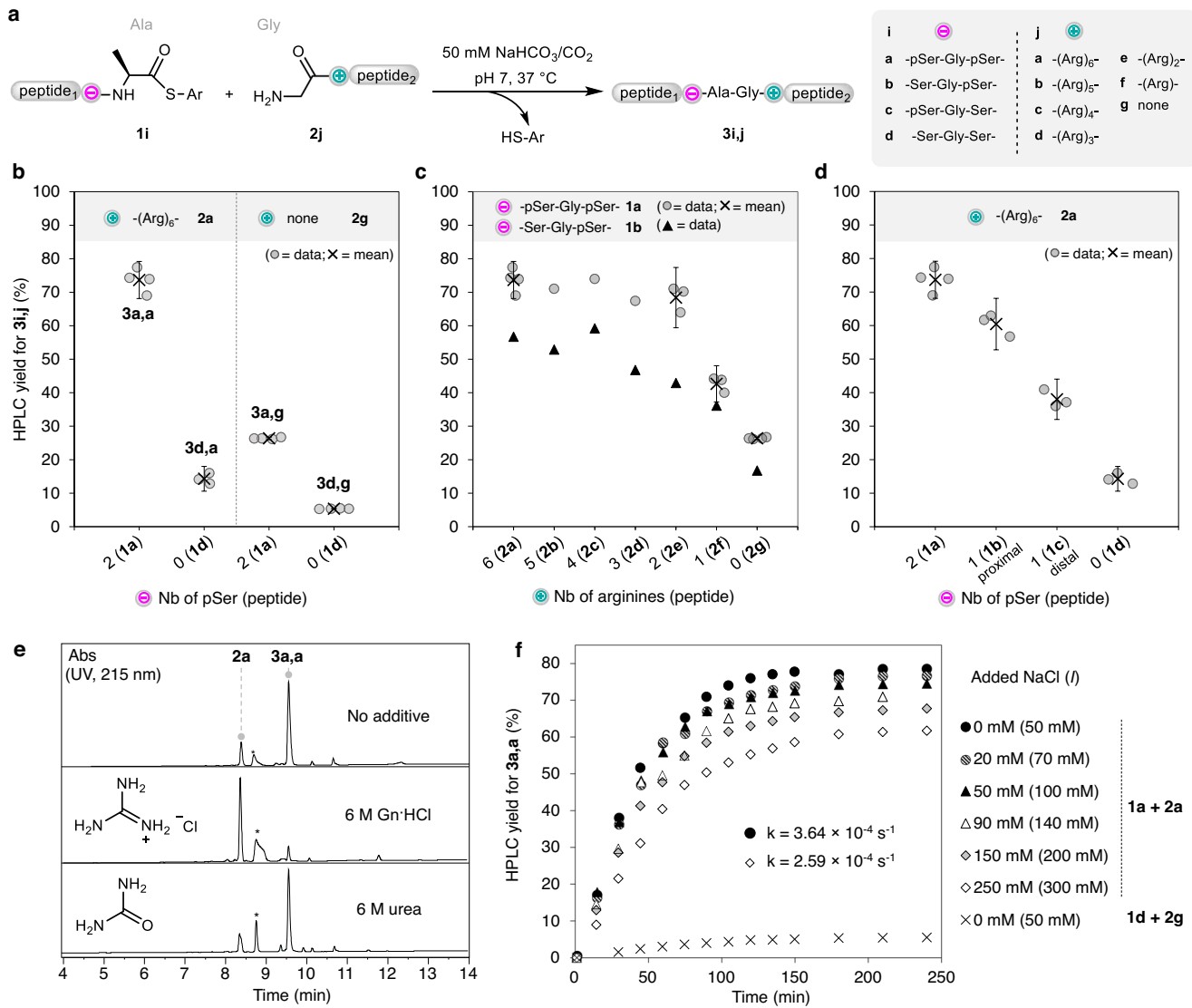

**Fig. 3 | Effect of module type and additives on the electrostatic assistance of peptide thioester aminolysis. a** Reactions studied. Conditions: Peptide thioester **1** (5 mM), glycyl peptide **2** (5 mM), sodium hydrogen carbonate (50 mM)/CO$_2$ buffer, 37 °C, pH 7. **b–d** When the error is indicated, the data correspond to the mean and standard error for independent experiments (95% confidence interval).
**b** Importance of the presence of complementary charged modules. **3a,a** $n = 4$, **3d,a** $n = 3$, **3a,g** $n = 4$, **3d,g** $n = 4$. **c** Effect of the number of Arg residues. **3a,a** $n = 4$, **3a,e** $n = 3$, **3a,f** $n = 3$, **3a,g** $n = 4$, otherwise, experiments were performed only once.

**d** Effect of the number and position of pSer residues. **3a,a** $n = 4$, **3b,a** $n = 3$, **3c,a** $n = 3$, **3d,a** $n = 3$. **e** Effect of additives (6 M Gn·HCl, 6 M urea, see Supplementary Methods for more data). LC-MS of the crude reaction mixtures after 17 h. *Peptide thioester hydrolysis byproduct. **f** The effect of the ionic strength was studied by adding NaCl ($I = 50$–300 mM). The data for the reaction of peptide thioester **1a** with glycyl peptide **2a** were fitted to a pseudo first-order rate law (see Supplementary Methods). The errors for rate constants correspond to the standard error of the non-linear regression analysis to a first-order rate law (95% confidence interval).

(**3a,a-e**). The data also indicate that a single arginine is sufficient to promote the reaction in comparison with the control without arginine. Regarding the negatively charged module, removing one phosphate group has a deleterious effect on the yield, especially for the position that is the closest to the thioester group (Fig. 3d). Logically, the aminolysis of peptide thioester **1b** with a single proximal pSer residue was found to be less tolerant than analogue **1a** to the decrease in the number of arginines, since in this case the yield decreased when the cationic module contained 4 arginines or less (Fig. 3c).

The yield of peptide **3a,a** from the reaction of peptide thioester **1a** with glycyl peptide **2a** was found to be almost independent of pH in the range 6–8 (see Supplementary Methods and Supplementary Fig. 173). In contrast, the yield of **3a,a** decreased sharply at pH <6, likely due to the concomitant protonation of the amine nucleophile (pK$_a$ ~ 8) and that of the phosphate groups (the pK$_a$ of pSer phosphate monoanion is ~5.6[31]). We also noticed that the aminolysis of peptide thioester **1a** by

glycyl peptide **2a** was inhibited by sodium phosphate (see Supplementary Figs. 174 and 175) or guanidinium chloride (Fig. 3e, see also Supplementary Figs. 176–178), bringing to the fore the key role played by phosphate-guanidinium interactions in the observed phenomena. Because guanidinium chloride is a well-known protein denaturant, we also tested the effect of urea, a nonionic protein denaturant. Our data show that the thioester aminolysis reaction is tolerant to 6 M urea (Fig. 3e and Supplementary Figs. 179–181). Taken together, these results show that the assistance proceeds exclusively through phosphate-guanidinium electrostatic interactions and does not involve any specific interaction between reactants. In agreement with the latter point, we were unable to detect any binding between peptides equipped with pSer–Gly–pSer and −(Arg)$_6$− modules by isothermal titration calorimetry.

Finally, we determined the apparent kinetic order of the aminolysis reaction. Surprisingly, the reaction of peptide thioester **1a** with

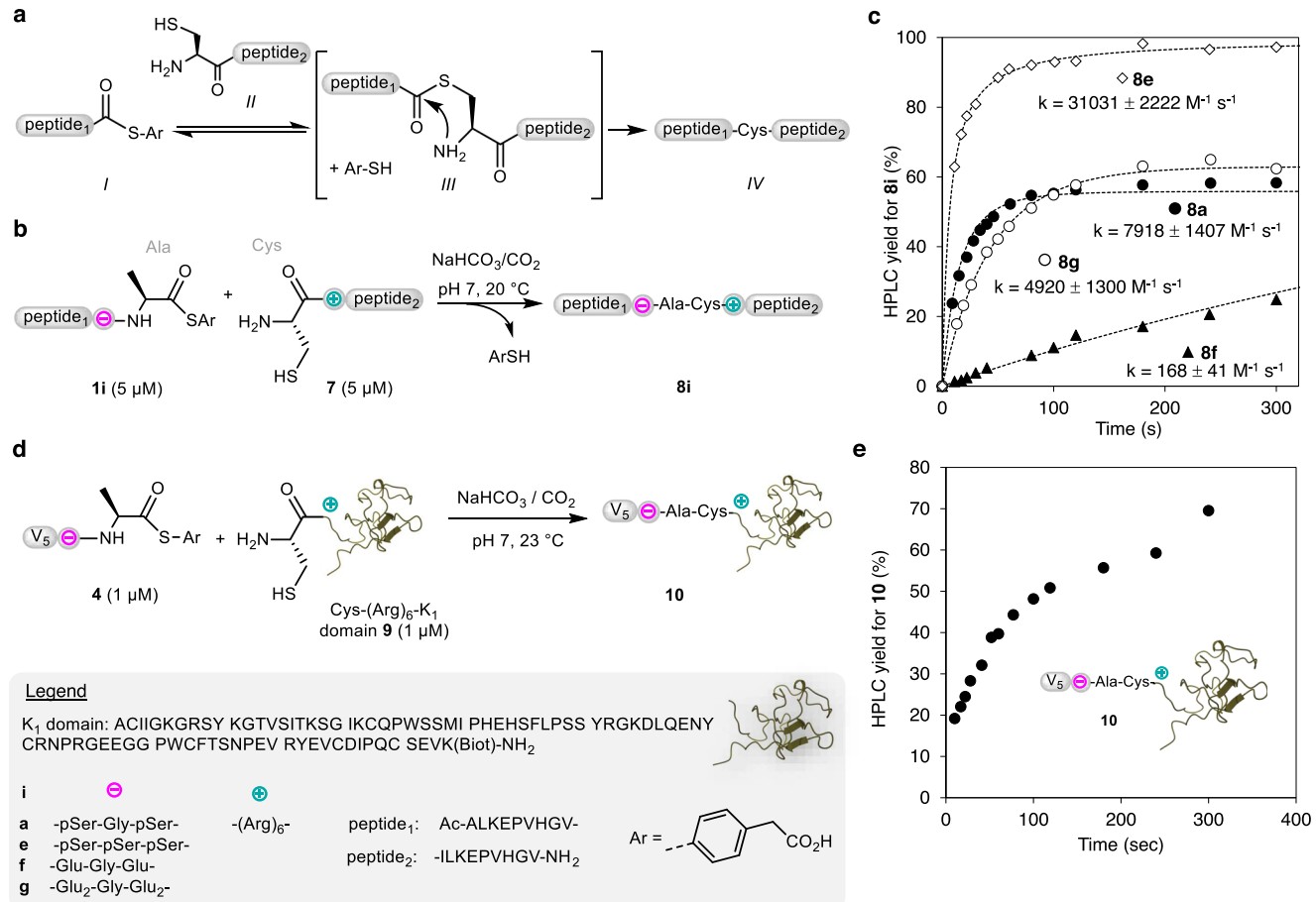

**Fig. 4 | Electrostatic assistance of the NCL reaction. a** Principle of the NCL reaction. **b, c** Assistance of the NCL reaction enables peptide formation in the low micromolar range with rates up to 31000 $M^{-1} s^{-1}$. The errors for rate constants correspond to the standard error generated by Kintek Global Kinetic Explorer Software during the numerical fitting of the data (95% confidence interval). **d, e** Application of electrostatic assistance to the modification of K1 HGF domain at 1 μM.

glycyl peptide **2a** followed a pseudo first-order rate law, as if the reaction was intramolecular (see Supplementary Fig. 197 and Supplementary table 1). Notably, increasing the ionic strength only slightly reduced the rate, and electrostatic assistance was still prominent at ionic strengths corresponding to physiological conditions ($I = 140$–200 mM) (Fig. 3f, see also Supplementary Methods, Supplementary Figs. 198–201, and Supplementary Table 2).

**Electrostatic assistance of the native chemical ligation**
Having demonstrated the power of phosphoserine-arginine electrostatic assistance in promoting the aminolysis of peptide thioesters, we looked at promoting the NCL reaction. NCL consists in reacting a peptide thioester **I** with a cysteinyl peptide **II** to produce a transient thioester-linked intermediate **III** (Fig. 4a). The latter spontaneously rearranges to give product **IV** having a peptide bond to cysteine. NCL is classically run at mM peptide concentrations and typically displays second-order rate constants of 0.3–4 $M^{-1} s^{-1}$ [32,33]. This reaction does not proceed at μM peptide concentrations unless it is assisted by a biomolecular template or other means [34,35].

The reaction of peptide thioester **1a** with cysteinyl peptide **7** was run at 5 μM and was found to proceed with a second-order rate constant of 7918 ± 1407 $M^{-1} s^{-1}$, about 3 orders of magnitude faster than non-assisted NCL (Fig. 4b, c). Part of the ligation product **8a** further evolved into a branched thioester by reacting with thioester **1a** through its internal Cys thiol (ligation product **8a** 58%, branched thioester byproduct ~17% yield, see Supplementary Methods, Supplementary Fig. 214 and Supplementary Table 3). No ligation product was

observed when peptide thioester **1a** was reacted with glycyl peptide **2a** at 5 μM, which lacks the side-chain Cys thiol, showing that product formation proceeds through an NCL mechanism as expected (see Supplementary Methods and Supplementary Fig. 217). Moreover, no ligation product was observed as well when peptide thioester **1a** was reacted with a Cys peptide lacking the Arg₆ module, i.e., CILKEPVHGV-NH₂, highlighting the critical role played by electrostatic assistance in this case too (see Supplementary Methods, and Supplementary Fig. 218). Importantly, ligation of the peptide thioester equipped with three phosphoserines, i.e., peptide thioester **1e**, with Cys peptide **7** proceeded at a rate of 31031 ± 2222 $M^{-1} s^{-1}$ (Fig. 4c). Such a five-digit rate constant, which is significantly higher than the one obtained with only two phosphoserines in the negatively charged module, enabled the ligation to be completed in less than 200 s at 5 μM, with yields as high as 97%. For comparison, one of the fastest NCL-like process designed so far, the diselenide-selenoester ligation, was shown to reach completion in 24 h at 5 μM [36,37]. In another example, a NCL reaction mediated by an oxalamide thioester surrogate was recently found to be complete in 7 h at the same peptide concentration [38].

When phosphoserines were replaced by glutamic acid residues used as negative charge carriers, ligations were successful as well (Fig. 4b, c, Supplementary Figs. 219 and 220, and Supplementary Tables 4 and 5). The rate achieved by using four glutamate residues (**1g** + **7** → **8g**, k = 4920 ± 1300 $M^{-1} s^{-1}$) which bring to the peptide thioester four negative charges as for the pSer–Gly–pSer module is significant. This is an important finding because contrary to pSer, Glu residue is genetically encoded. Extra Glu residues can thus be easily

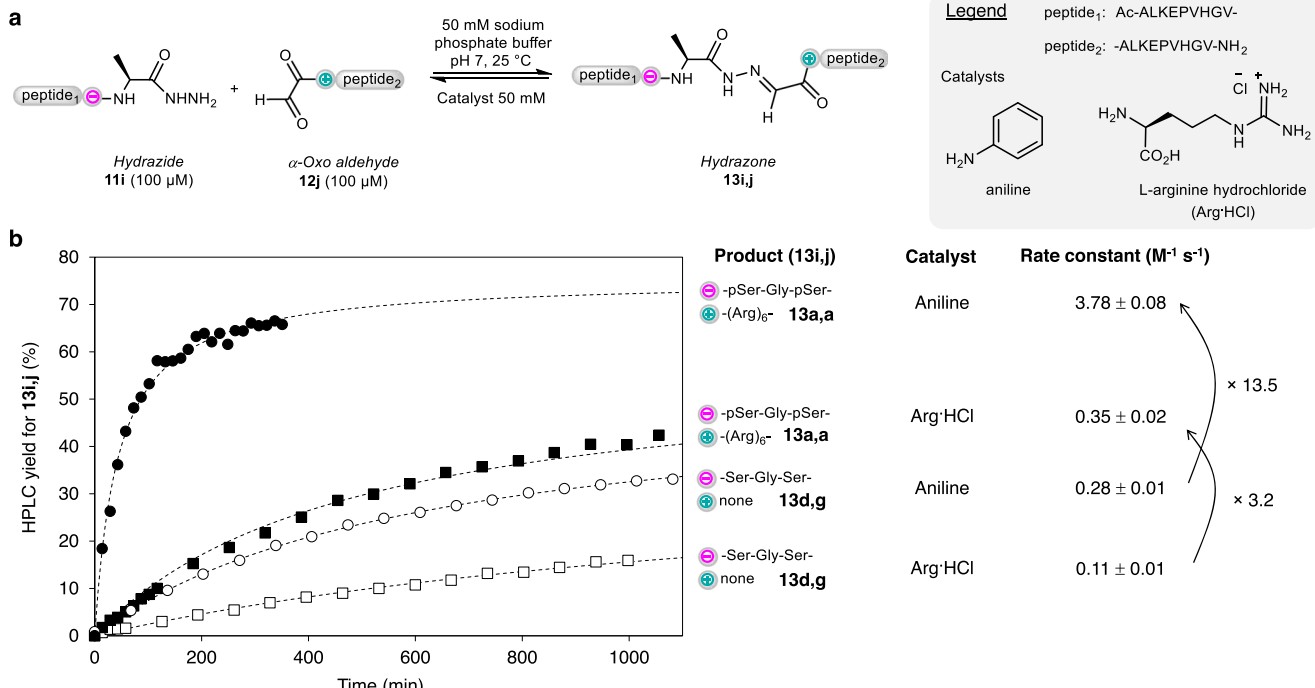

**Fig. 5 | Electrostatic assistance of hydrazone ligation. a** Principle. **b** kinetic data and fitting curves (dashed lines) of hydrazone ligations conducted in the presence of arginine hydrochloride (Arg·HCl) or aniline as nucleophilic catalysts. The reactions were performed once. The rate constant error corresponds to the standard error of the fit (95% confidence interval).

incorporated into proteins using the cell machinery. Nevertheless, the rate achieved with four Glu residues is below that obtained with the two-phosphoserine-based module **1a** (**1a** + **7** → **8a**, k = 7918 ± 1407 M$^{-1}$ s$^{-1}$). Although this experiment shows the importance of module overall charge on electrostatic assistance, it reveals that a phosphoserine residue cannot be fully recapitulated by two glutamic acid residues as already noticed by others.

An important question that we also investigated is whether the presence of negatively charged amino acid residues near the positively charged module might abolish the long-range electrostatic attraction mechanism either by competing intramolecularly with the negatively charged module present on the ligation partner, or by repelling it. To investigate this point, we produced two variants of Cys peptide **7** by extending its C-terminus with pSer–Gly–pSer or Glu–Glu–Gly–Glu–Glu sequences. The reaction of these Cys peptides with peptide thioester **1a** equipped with pSer–Gly–pSer module proceeded similarly according to an apparent second-order rate constant of ~700 M$^{-1}$ s$^{-1}$ (see Supplementary Methods, Supplementary Fig. 221, and Supplementary Table 6). We conclude that although the presence of negatively charged amino acid residues near the positively charged module affects the rate of ligation, electrostatic assistance remains nevertheless significant in this case too.

The application of electrostatic assistance for promoting the modification of a 10 kDa polypeptide derived from the kringle 1 (K1) domain of hepatocyte growth factor by NCL is shown in Fig. 4d. The reaction of Cys polypeptide **9** with peptide thioester **4** (pSer–Gly–pSer module) was completed in a few minutes at 1 μM and the yield for ligation product **10** amounted to ~70% (Fig. 4e). We noted that branched thioester formation was insignificant in this case. Similarly, the reaction of Cys polypeptide **9** with peptide thioester **1g** (Glu-Glu-Gly-Glu-Glu module) proved successful (see Supplementary Methods and Supplementary Figs. 226 and 227). Non-ionic detergent *n*-octyl glucoside, which was used as an additive in these experiments to prevent the binding of K1 polypeptide to the plastic tube used for the reaction, does not affect the electrostatic

assistance (see Supplementary Methods and Supplementary Figs. 184 and 185).

## Beyond peptide bond-forming reactions

The last question we investigated is whether phosphoserine–arginine electrostatic assistance can efficiently assist reactions mechanistically unrelated to thioester aminolysis. We showed that this is effectively the case by studying the rate of hydrazone ligation using peptide hydrazides and α-oxo aldehyde peptides as reactants. The hydrazone-forming reaction studied in this work is depicted in Fig. 5a. The nucleophile (peptide hydrazide **11**) is associated with the negatively charged module, while the electrophile (peptide α-oxo aldehyde **12**) carries the positively charged module. For the thioester aminolysis and NCL reactions, the nucleophile (Gly or Cys) was associated with the positively charged module. Therefore, compared to the peptide bond-forming reactions described in the previous sections, the position of the nucleophile/electrophile relative to the charged modules is inverted. Another important difference is that hydrazone ligation is an equilibrated reaction, while thioester aminolysis/NCL reactions are irreversible processes.

The reactions were conducted in phosphate buffer (50 mM) at neutral pH, in the presence of aniline[26,27,39] or arginine[40] acting as nucleophilic catalysts. The results of the kinetic study are presented in Fig. 5b. The kinetic data could be fitted to an apparent second-order rate law (dashed lines). Based on the obtained rate constants, hydrazone formation proceeds ~3 times faster with the peptide reactants equipped with the charged modules (**13a,a** vs **13d,g**) when the catalyst is arginine (50 mM)[40], and 13 times faster with aniline catalyst used at the same concentration. Another series of experiments were performed using a larger diversity of charged peptide modules and arginine as catalyst to enable the monitoring of hydrazone formation by UV spectroscopy (see Supplementary Methods, Supplementary Figs. 228–230, and Supplementary Table 7). The taken message from this study is that electrostatic assistance synergizes with amine catalysts to promote hydrazone formation.

## Discussion

One hallmark of the work reported here is the simplicity by which rate accelerations by up to several orders of magnitude can be achieved. Indeed, equipping the reactants with oppositely charged modules made of only a few residues (phosphoserine, glutamic acid or arginine) is sufficient to significantly impact the rate of chemical reactions at neutral pH. Regarding the negative charge carrier, phosphoserine provided the highest rate enhancements, which could not be recapitulated by a glutamic acid dyad of the same charge, i.e., Glu-Glu. This is likely due to the charge density of the doubly charged phosphate ester group, which makes it more effective for attracting cations than two singly charged carboxylate groups separated by a certain distance. Another hallmark of the charged modules studied in this work is their effectiveness under denaturing conditions, and the lack of binding through isothermal titration calorimetry experiments. We cannot definitively rule out the occurrence of a weak affinity between the charged modules, considering the very weak binding constant of arginine to phosphate ion in bulk water ($\sim 0.1\,M^{-1}$)[41]. However, the fact that the oppositely charged modules do not form a stable binding interface is not unexpected since the formation of salt bridges would occur at a high dehydration penalty[42]. In such a case for a stable and well-defined interface to form, the dehydration penalty must be compensated by short-range interactions and this is typically what happens when two proteins form their high-affinity binding interface after encounter[43]. Such mechanisms are unlikely for the peptide modules studied in this work due to their small size. Rather, the increased reaction rates observed in this work can possibly be accounted by a higher frequency of collision when the peptide reactants are equipped with the charged modules.

Although the intimate mechanisms implicated in rate enhancements can be certainly debated, they seem to be widely applicable. The effect was observed for three different chemical reactions, although they proceed by distinct mechanisms. Significant rate accelerations were achieved using amine or thiolate nucleophiles in reaction with peptide thioesters, i.e., for the peptide thioester aminolysis and NCL reactions, respectively. Both processes are irreversible and lead to the formation of a peptide bond. Note that the peptide thioester aminolysis reaction is not selective for the N-terminus of proteins, but that it becomes selective with electrostatic assistance. The method was also useful for promoting hydrazone ligation, which is an equilibrated process. Interestingly in this case, electrostatic assistance and nucleophilic catalysis by amine compounds were found to synergize. The combination of these two assistance mechanisms should broaden the scope of hydrazone ligation. One immediate advantage of implementing electrostatic assistance for such a reaction is the possibility to achieve significant rates enhancements while using less of a toxic amine catalyst such as aniline. Using the thioester aminolysis reaction, we showed that the electrostatic assistance is also operative under ionic strengths corresponding to physiological conditions. Therefore, the method can potentially enlarge the scope of many chemical transformations used for protein modification. The position and number of charged residues can be adjusted in each case to achieve the desired degree of assistance. Last but not least, the cationic module made of arginines and the negative module made of glutamic acid residues can easily be programmed in live cells. The use of protein reactants of biological origin enables the application of the method to the semi-synthesis of complex protein scaffolds.

To conclude, we have shown that a mechanism used by cells to promote the rate of association of proteins to other biomolecules, i.e., the long-range electrostatic attraction of oppositely charged amino acid residues, can be replicated in a chemical reactor to assist and orient the formation of a covalent bond between peptide and protein reactants. In addition to enlarging the scope of well-known selective chemical transformations used for N-terminal protein modification, the approach might also restore interest in reactions that suffer from

being non-site selective, as has been done in this work with the peptide thioester aminolysis reaction.

## Methods

### Preparation of sodium bicarbonate/$CO_2$ buffer

**For peptide thioester aminolysis.** Sodium hydrogen carbonate (21.0 mg, 0.250 mmol, 50 mM) was dissolved in water (5 mL) in an opened flask. The solution was placed in a $CO_2$ incubator for cell biology (5% partial $CO_2$ pressure, 37 °C, water-saturated) during 24 h. The pH of the equilibrated solution was pH ~8.30.

### Synthesis of peptide hydrazides

Peptide hydrazides were used as precursors of peptide thioesters[22] or as reactants for the study of hydrazone ligation.

**Preparation of hydrazine solid support.** Hydrazine solid support was prepared by adapting known protocols[22,44]. For a 0.1 mmol scale synthesis, 0.3 mmol of 2-chlorotrityl chloride 1% divinylbenzene cross-linked polystyrene (2-CTC solid support, 0.6 mmol g⁻¹) was swelled in *N,N*-dimethylformamide (DMF, 2.5 mL). After 15 min, DMF was drained and the solid support was cooled at 0 °C. A solution of hydrazine hydrate (4.00 equiv, 1.20 mmol, 76.8 μL) and triethylamine (6.00 equiv, 1.80 mmol, 251 μL) in DMF (1 mL) was added slowly to the beads at 0 °C. The bead suspension was agitated for 1 h at rt. The reaction was quenched by adding methanol (80.3 μL) to the bead suspension, which was further agitated for 10 min. The beads were then washed with DMF (2 × 2 min), water (2 × 2 min) and DMF (3 × 2 min).

**Coupling of the first amino acid.** The first amino acid (4 equiv) was coupled using 2-(1*H*-benzotriazol-1-yl)-1,1,3,3-tetramethyluronium hexafluorophosphate (HBTU, 3.8 equiv)/*N,N*-diisopropylethylamine (DIEA, 8 equiv) activation in DMF. The amino acid was pre-activated for 30 s at room temperature (rt) before being added to the beads. The bead suspension was agitated for 1 h, and then washed with DMF (2 × 2 min). The coupling was repeated and the beads were washed with DMF (2 × 2 min), dichloromethane (DCM, 6 × 2 min) and MeOH (2 × 2 min). The beads were finally dried in vacuo for 2 h. The loading was determined by treating aliquots with piperidine in DMF (20% v/v) and measuring the absorbance of the dibenzofulvene-piperidine adduct at 280 nm. Loading 0.20 mmol g⁻¹.

**Coupling of Fmoc-Ser(PO(OBzl)OH)-OH.** Fmoc-Ser(PO(OBzl)OH)-OH (5 equiv) was coupled manually using 2-(1*H*-benzotriazol-1-yl)-1,1,3,3-tetramethylaminium tetrafluoroborate (TBTU, 5 equiv)/1-hydroxybenzotriazole (HOBt, 5 equiv)/DIEA (15 equiv) activation in DMF. The reagents were mixed and the solution was immediately added to the peptidyl resin. The bead suspension was agitated for 2 h, and then washed with DMF (3 × 1 min), acetylated using Ac₂O/DIEA/DMF 10/5/85 v/v/v and washed with DMF (3 × 1 min) and DCM (3 × 1 min).

**Peptide elongation and cleavage.** Peptide elongation and cleavage were performed using standard 9-fluorenylmethoxycarbonyl (Fmoc) solid phase peptide synthesis (SPPS) as described in the Supplementary Methods.

### Synthesis of the peptide thioesters

**General procedure.** The peptide hydrazide (11.4 μmol, 3 mM final concentration) was dissolved in 0.2 M phosphate buffer containing 6 M Gn•HCl, pH = 3 (1.899 mL) and cooled to −14 °C using a NaCl/ice bath. NaNO₂ (10 equiv, 0.5 M in water, 228 μL) was added to the peptide, and the reaction mixture was agitated for 15 min. Then, 4-mercaptophenylacetic acid (MPAA, 100 equiv, solution in 0.2 M phosphate buffer containing 6 M Gn•HCl, pH = 6.5, 1.889 mL) was added to the reaction mixture. The NaCl/ice bath was removed, and

the pH of the reaction mixture was adjusted to pH 6.8. After 15–20 min, glacial acetic acid was added (10% v/v, 380 μL) and the aqueous solution extracted with $Et_2O$ (10×) to remove the excess of MPAA. The crude peptide thioester was purified by RP-HPLC using a C18 XBridge column (50 °C, 215 nm, 6 mL min$^{-1}$, eluent A = water containing 0.1% v/v of TFA, eluent B = $CH_3CN$ containing 0.1% v/v of TFA). The gradient used for the purification is specified for each peptide thioester in the Supplementary Methods. The purified fractions were pooled, frozen and lyophilized.

## Analysis of kinetic data

Conversion to ligated product was calculated from the UV trace of the chromatograms, at a wavelength of 215 nm. Prior to kinetic data analysis and fitting, conversions were normalized and transformed into concentrations of the product based on the starting aryl thioester peptide 1a concentration.

Kintek Global Kinetic Explorer Software, Version 10.0.200514, was used for kinetic modeling. The standard deviation for each trace was first estimated upon fitting the experimental dataset with an analytical function (3-exponential) to determine an average sigma value, further used for numerical data fitting. The subsequent numerical fit allowed determining the apparent rate constant $k_{app}$ for each experiment. Fitting to a given model was achieved by nonlinear regression analysis based upon an iterative search to find a set of reaction parameters that gives a minimum $\chi^2$. The process was completed by careful visual examination of the fits.

Peptide thioester aminolysis reactions were fitted to a pseudo-first-order rate law, while NCL and hydrazone ligation reactions were fitted to a second-order rate law. For experiments performed once, rate constants are presented as value ± standard error of the fit (95% confidence interval). For experiments replicated three times or more, rate constants are presented as mean ± standard error (95% confidence interval).

## Peptide thioester aminolysis reaction (Fig. 2)

**General procedure.** A typical experimental procedure is illustrated with the synthesis of peptide 3a,a by reaction of peptide thioester 1a with glycyl peptide 2a.

Glycyl peptide GRRRRRRALKEPVHGV-NH$_2$ 2a (6.60 mg, 2.22 μmol, 5 mM final concentration) was dissolved in 50 mM sodium bicarbonate/$CO_2$ buffer (444 μL). This solution was then added to the 0.6 mL plastic tube containing peptide thioester Ac-ALKEPVHGVpSGpSA-MPAA 1a (1 equiv, 4.07 mg, 5 mM final concentration). The solution was vortexed a few seconds and placed in a $CO_2$ incubator for cell biology (5% $CO_2$ partial pressure, 37 °C, water-saturated atmosphere). A needle was inserted through the cap to allow $CO_2$ to diffuse into the tube and the reaction mixture. The final pH of the reaction mixture was 7.11.

After 24 h, the reaction mixture was analyzed by LC-MS (Supplementary Fig. 186). Then, the reaction mixture was diluted with 10% aqueous AcOH (10% by vol, 3.5 mL). Diethyl ether extractions (three times) were done to extract MPAA before HPLC purification. The RP-HPLC purification was performed using a C18 XBridge column (50 °C, 215 nm, 6 mL min$^{-1}$, eluent A = water containing 0.1% v/v of TFA, eluent B = $CH_3CN$ containing 0.1% v/v of TFA, 0 to 5% B in 3 min, then 5 to 25% B in 40 min) and furnished 6.12 mg (61%) of purified peptide 3a,a. The characterization of purified peptide 3a,a is presented in Supplementary Figs. 187 and 188. A reference peptide was produced by classical SPPS (Supplementary Figs. 189 and 190). The co-injection of peptide 3a,a with the reference peptide by UPLC-MS is shown in Supplementary Figs. 191 and 192.

A sample was sent to C.A.T. GmbH (Tübingen, Deutschland) to determine the extent of D-Ala in the peptide by chiral GC-MS (0.97% D-Ala).

## Synthesis of titin conjugate 6 (Fig. 2)

GR$_6$-I$_{27}$ titin protein 5 (2.59 mg, 0.2 μmol, 0.8 mM final concentration) was dissolved in a solution composed of 1 mM TCEP, 10 mM n-octylglucoside in 20 mM sodium bicarbonate/$CO_2$ buffer (250 μL). The solution was transferred to the 0.6 mL low-bind plastic tube containing Ac-V5-pSGpSA-MPAA peptide 4 (1.251 mg, 0.4 μmol, 2 equiv, 1.6 mM final concentration). The reaction mixture was turbid. The solution was vortexed a few seconds and placed in a $CO_2$ incubator for cell biology (5% $CO_2$ partial pressure, 37 °C, water-saturated) with a needle inserted through the cap to allow $CO_2$ to diffuse into the tube. The final pH of the reaction mixture was 6.27. After 20 h, solid Gn•HCl (143.3 mg, ~3 M final concentration) was added to the reaction mixture. Then, solid TCEP • HCl (10 equiv, 2 μmol, 573 μg) was added and pH was adjusted to 7.09 to reduce mixed disulfides observed by LC-MS. After 30 min, the reaction mixture was acidified by adding AcOH (50 μL) and extracted 3 times with diethyl ether. The crude mixture was purified by HPLC using a C3 zorbax column (70 °C, 215 nm, 6 mL min$^{-1}$, eluent A = water containing 0.1% v/v of TFA, eluent B = $CH_3CN$ containing 0.1% v/v of TFA, 0 to 20% B in 5 min, then 20 to 40% B in 35 min) to give 614 μg of conjugate 6 (20%). The characterization of conjugate 6 is provided in Supplementary Figs. 204–210.

## Factors influencing the electrostatic assistance of peptide thioester aminolysis (Fig. 3)

**General protocol.** A typical experimental procedure for monitoring the peptide thioester aminolysis reaction on the analytical scale is given for the reaction of peptide thioester 1a with glycyl peptide 2a.

GRRRRRRALKEPVHGV-NH$_2$ peptide 2a (~0.40 μmol, 5 mM final concentration) was dissolved in 50 mM sodium bicarbonate/$CO_2$ buffer (~80 μL). The solution was then transferred to the 0.6 mL plastic tube containing Ac-ALKEPVHGVpSGpSA-MPAA peptide thioester 1a (1 equiv, 5 mM final concentration). The solution was vortexed a few seconds and placed in a $CO_2$ incubator for cell biology (5% $CO_2$ partial pressure, 37 °C, water saturated atmosphere). A needle was inserted through the cap to allow $CO_2$ to diffuse into the tube and the reaction mixture. The final pH of the reaction mixture was 7.13.

The mixture was analyzed by LC-MS after diluting an aliquot (1 μL) with aqueous acetic acid (10% AcOH in water, 100 μL).

## Electrostatic assistance of the native chemical ligation (Fig. 4)

**Typical experimental procedure (Fig. 4b, c).** The reaction was performed at 20 °C. Sodium bicarbonate (16.80 mg, 20 mM final concentration) was dissolved in water (10 mL).

A 50 mL plastic tube equipped with a magnetic bar was filled with this solution (10 mL). A pH electrode was immersed in the solution to continuously measure the pH. The pH was regulated by bubbling $CO_2$ into the solution under stirring (430 rpm). The flow of $CO_2$ into the solution was controlled by using a syringe driver equipped with a 50-mL syringe filled with $CO_2$. The flow rate of $CO_2$ was around 1 mL min$^{-1}$.

Once the pH was stable, CRRRRRRALKEPVHGV-NH$_2$ peptide 7 (1 μL of a 50 mM stock solution in water, $5 \times 10^{-8}$ mol, 5 μM final concentration) was added under stirring (430 rpm) to the above aqueous solution in the plastic tube. A few seconds later, the peptide thioester Ac-ALKEPVHGVpSGpSA-MPAA 1a (1 μL of a 50 mM stock solution in water, $5.10^{-8}$ mol, 5 μM final concentration) was added to the mixture under stirring (430 rpm). The reaction was analyzed by UPLC-MS after quenching an aliquot of the reaction mixture (100 μL) with glacial acetic acid (20 μL). A typical UPLC-MS analysis can be found in Supplementary Figs. 212 and 213. The time course of the ligation and branched byproduct formation is presented in Supplementary Figs. 211 and 214.

**Reproducibility.** The reaction of peptide thioester 1a with cysteinyl peptide 7 at pH 7 was repeated three times to show the reproducibility

of the assisted NCL process. The data for these experiments are shown in Supplementary Fig. 215.

### Synthesis of conjugate 10 (Fig. 4)

The procedure used to produce conjugate **10** is similar to the one used for performing NCL at 5 μM with some adaptations.

The reaction was performed at 20 °C. Sodium bicarbonate (5.04 mg, 20 mM final concentration) was dissolved in water (3 mL). The ligation was done at 1 μM protein concentration using a 5 mL low-bind plastic tube equipped with a magnetic stirrer (600 rpm). The plastic tube was filled with the sodium bicarbonate solution (3 mL), which was supplemented with TCEP • HCl (846 μg, 1 mM final concentration) and *n*-octylglucoside (8.63 mg, 10 mM final concentration). The latter additive was used to prevent the precipitation of the proteins or their binding to the plastic tube.

The flow of $CO_2$ was adjusted to obtain a pH of 6.9–7.0. $CR_6$-K1 protein **9** (38 μg, 2.9 nmol, 1 μM final concentration) was added under stirring (600 rpm) to the buffer in the plastic tube. A few seconds later, the peptide thioester **4** (0.29 μL of a 10 mM stock solution in water, 2.9 nmol, 1 μM final concentration) was added to the mixture under stirring (600 rpm).

The reaction mixture was monitored by RP-HPLC and LC-MS after quenching an aliquot (100 μL) of the reaction mixture with glacial acetic acid (20 μL). The data obtained after a reaction time of 5 min are presented in Supplementary Figs. 222 and 223. The reaction mixture was also analyzed by SDS-PAGE using streptavidin-horseradish peroxidase, see Supplementary Methods and Supplementary Figs. 224 and 225.

### Electrostatic assistance of hydrazone ligation

Two types of hydrazone ligation studies were performed. In the first series of experiments described in Fig. 5, hydrazone formation catalyzed either by Arg•HCl or aniline was monitored by UPLC-MS with a limited number of peptide reactants.

In the second series of experiments, hydrazone ligation catalyzed by Arg•HCl was monitored by UV (see Supplementary Methods, Supplementary Fig. 229, and Supplementary Table 7). With such an assay, we could enlarge the type of peptide reactants used to form the hydrazone. It is not compatible with aniline, which strongly absorbs UV light at the wavelength used for monitoring product formation.

**Typical experimental procedure**. Stock solutions (10 mM) of peptide hydrazide Ac-ALKEPVHGVSGSA-NHNH₂ **11i** and peptide aldehyde **12j** were diluted in the appropriate buffer (0.1 mM final concentration for each peptide, pH 7.06, 25 °C). The peptide aldehyde **12j** (1 μL) was added first in the buffer (98 μL) introduced in a glass tube. The glass tube was vortexed and centrifuged. Then, the peptide hydrazide (1 μL) was added. The tube was vortexed and centrifuged again. The reaction mixture was placed in the sample chamber of the UPLC system maintained at 25 °C. The reaction mixture was analyzed by UPLC-MS (10 μL of the reaction mixture was injected directly on a 300 SB C3 column 2.1 × 100 mm, 1.8 μm, 50 °C, 215 nm, 0.4 mL min⁻¹, eluent A = water containing 0.1% v/v of TFA, eluent B = $CH_3CN$ containing 0.1% v/v of TFA, 14 to 44% B in 5 min).

**Remark**. The UPLC-MS gradient was carefully optimized to avoid premature hydrazone formation during analysis. The gradient used in this experiment might not be appropriate for other peptide sequences and should be adapted in each case. An important control experiment to verify that analysis does not generate a bias in the determination of hydrazone yields is to inject the reaction mixture immediately after having mixed the peptide reactants. The UPLC analysis of such a mixture should show no hydrazone formation. Analysis by classical HPLC was always found to generate strong biases in our hands.

### Reporting summary

Further information on research design is available in the Nature Research Reporting Summary linked to this article.

## Data availability

The data generated in this study are provided in the article, Supplementary Information, and Source Data file. Data are available from the corresponding author upon request. Source data are provided with this paper.

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

## Acknowledgements
We acknowledge financial support from CNRS, the University of Lille, Pasteur Institute of Lille, and INSERM. We thank Hervé Drobecq for performing proteomic analyses.

## Author contributions
O.M. conceptualized and supervised the study and wrote the manuscript. V.A. supervised the study, performed nonlinear regression analyses of kinetic data, and wrote the manuscript. N.O. performed the experimental work and wrote the manuscript. R.D., M.S., and B.S. performed the experimental work.

## Competing interests
The authors declare no competing interests.
