## [Peer Review File · Nature Communications]

A biomimetic electrostatic assistance for guiding and promoting N-terminal protein chemical modificationREVIEWER COMMENTS

Reviewer #1 (Remarks to the Author):

Electrostatics can mediate the binding of protein/protein, protein/DNA and others in living cells. The mechanism used by cells was employed in synthetic organic chemistry to assist the formation of covalent bond between small molecules. However, the electrostatic catalysis to promote the formation of a covalent bond between protein reactants remain challenging. In this manuscript, the authors described a biomimetic electrostatic catalysis strategy to promote chemical protein modification. The key feature of the new method is introducing a negatively charged phosphoserines (PSer-Gly-pSer) in one reactant and an oppositely charged arginines (Arg6) in the other reactant to accelerating the chemical modification of proteins. There is no detectable affinity between the charged reactants. Moreover, the electrostatics-assisted strategy enable the native chemical ligation with three orders of magnitude to proceed peptide ligation in the low micromolar range. The practicality of the method was demonstrated by the modification of K1 HGF domain at 1 μM . The new method paves a new way for protein modification and semi-synthesis of complex protein scaffolds in live cells. Overall, the work should be published in Nat Commun after addressing these issues.

Comments:

1. If the pSer-Gly-pSer-tagged peptide contains several Args, does the electrostatic interaction-mediated modification still work well or can be self-inhibited?
2. The authors should explain why use OG in the reaction between 4 and 5.
3. Why the modification reaction should be performed under the carbon dioxide condition?
4. How about the reaction if there is three or four pSer in the peptide for ligation in figure 3?
5. Please delete one "." in line 6 at page 7.

Reviewer #2 (Remarks to the Author):

Ollivier et al. describe the application of electrostatic catalysis to the area of peptide and protein conjugation reactions. The underlying concept of artificially increasing affinity between two reaction partners by positive electrostatic interactions is not new and has seen a certain revival over the past years. Here, the authors take advantage of the bi-dentate interaction of phosphate (on phosphoserine residues) introduced close to the conjugation site on one reaction partner with multiple arginine residues its counterpart. This provides sufficient affinity to bring reactions partners together and to achieve selective reactions between two peptides/proteins at the desired N-terminal amino group of a protein domain and an synthetic peptide carrying a C-terminal thioester in a direct aminolysis reaction. Such reactions have been pioneered by Aimoto et al. for linking synthetic peptides via a native amide bond (this should be referenced in this manuscript) and proceed at very high concentrations only. They also require protection of sidechain amino groups of lysine residues, which is not needed here. In addition the authors nicely demonstrate the pseudo-first order reaction kinetics, most likely due to formation of a pre-ligation complex based on electrostatic interactions. Addition of salt such as phosphate of the charged chaotrope Gnd-HCl indicates that in deed electrostatic interactions are facilitating these reactions. The required number of Arg and pSer residues is also analyzed. Furthermore the reaction is extended to native chemical ligation and shows a remarkable increase in reaction kinetics at very low concentrations (1-5 μM), which are typically not within the range of standard NCL reaction (but can be used in DSL and EPSL reaction that should be mentioned as well). Another example is hydrazine formation even though no data is provided in the main manuscript.

Overall, I really appreciate the very solid study on the use of electrostatic catalysis in peptide and protein ligation and its potential applications. I believe this manuscript could be suited for publication in Nat Comm after the authors have addressed several points:

The title refers to "protein chemical modification", however the data shown is really on ligation

reactions as the electrostatic tags are always placed at the N-or C-termini. Please adjust the title accordingly.

The conclusions refers to the fact that the Arg-tag can be genetically encoded, opening the way for live cell modifications. However, the authors need to critically discuss that the pSer are not genetically encoded and that they are introduced by chemical synthesis.

This also brings up the question why the authors have not tested Asp or Glu to induce the electrostatic interaction (as these are genetically encoded)? Ideally an example with an Asp or Glu tag would be included in this study.

A more generally important point is the discussion of the pSer and Arg residues needed at the ligation site. Many peptide/protein ligations are setup to avoid such ligation scars, especially such highly charged ones that can easily impact protein folding/function. This needs to be discussed in the manuscript.

Here, options of releasable tags should be considered.

What is the authors opinion of their reactions with highly charged proteins? A nice control would be a test reaction with peptides containing additional pSer or Arg residues further away from the reactive termini.

The acceleration of NCL by including pSer and Arg tags is impressive but formation of the branched product is a concern. Shouldn't electrostatic repulsion of pSer reduce this side product?

The data on hydrazone formation from a peptide with C-terminal hydrazide and one with N-terminal aldehyde is hidden in the SI. I believe this needs to be included in the main MS and rates should also be compared to aniline-catalyzed ligation reactions.

Dear reviewers,

We really appreciate the comments and suggestions made.

We took the time to perform additional studies to respond to all your questions, and this really strengthens our manuscript. Our response to our comments is inserted within your reports below (in blue).

Best Regards

Oleg Melnyk

REVIEWER COMMENTS

Reviewer #1 (Remarks to the Author):

Electrostatics can mediate the binding of protein/protein, protein/DNA and others in living cells. The mechanism used by cells was employed in synthetic organic chemistry to assist the formation of covalent bond between small molecules. However, the electrostatic catalysis to promote the formation of a covalent bond between protein reactants remain challenging. In this manuscript, the authors described a biomimetic electrostatic catalysis strategy to promote chemical protein modification. The key feature of the new method is introducing a negatively charged phosphoserines (PSer-Gly-pSer) in one reactant and an oppositely charged arginines (Arg6) in the other reactant to accelerating the chemical modification of proteins. There is no detectable affinity between the charged reactants. Moreover, the electrostatics-assisted strategy enable the native chemical ligation with three orders of magnitude to proceed peptide ligation in the low micromolar range. The practicality of the methode was demonstrated by the modification of K1 HGF domain at 1 μ M. The new method paves a new way for protein modification and semi-synthesis of complex protein scaffolds in live cells. Overall, the work should be published in Nat Commun after addressing these issues.

Comments:

1. If the pSer-Gly-pSer-tagged peptide contains several Args, does the electrostatic interaction-mediated modification still work well or can be self-inhibited?

This is an interesting question that we investigated using the NCL reaction and by placing negatively charged residues near the positively charged module. We did that because Cys peptides having such dual modifications are easier to prepare than modified peptide thioesters as originally suggested by this reviewer. To peptide were prepared to evaluate the impact of introducing pSer and Glu residues.

The following paragraph was inserted in the section "*Electrostatic catalysis of the native chemical ligation*" to discuss this point:

An important question that we also investigated is whether the presence of negatively charged AAs near the positively charged module might abolish the long-range electrostatic attraction mechanism

either by competing intramolecularly with the negatively charged module present on the ligation partner, or by repelling it. To investigate this point, we produced two variants of Cys peptide **7** by extending its C-terminus with pSer-Gly-pSer or Glu-Glu-Gly-Glu-Glu sequences. The reaction of these Cys peptides with peptide thioester **1a** equipped with pSer-Gly-pSer module proceeded similarly according an apparent second order rate constant of $\sim 700 \text{ M}^{-1} \text{ s}^{-1}$ (see Supplementary Methods). We conclude that although the presence of negatively charged AAs near the positively charged module affects the rate of ligation, electrostatic catalysis remains nevertheless significant in this case too.

The Supplementary Information was modified accordingly (preparation of the peptide reactants and results of the ligation experiments).

2. The authors should explain why use OG in the reaction between 4 and 5.

We added this sentence at the end of the paragraph discussing the application of electrostatic catalysis to the modification of proteins using NCL, in the section « *Electrostatic catalysis of the native chemical ligation* »:

Non-ionic detergent *n*-octyl glucoside (OG), which was used as an additive in these experiments to prevent the binding of K1 polypeptide to the plastic tube, does not affect the electrostatic assistance (see Supplementary Methods).

3. Why the modification reaction be performed under the carbon dioxide condition?

We added in the section “*Proof-of-concept for electrostatic catalysis at the protein level* »:

Regarding the buffer used for performing the reaction and although 50 mM sodium phosphate is well tolerated, we preferred to use sodium hydrogen carbonate/carbon dioxide as the buffering system because sodium phosphate buffer, which is more classically used for conducting peptide ligation reactions, competes with the phosphoserine residues (see Supplementary Methods). Moreover, phosphate buffers can promote the epimerization and hydrolysis of peptide aryl thioesters of the type used in this work.¹³

4. How about the reaction if there is three or four pSer in the peptide for ligation in figure 3?

We thank the reviewer for this suggestion. We took the decision to test a triphosphoserine module using the native chemical ligation for its broad scope and not the thioester aminolysis reaction as proposed by the reviewer.

In this case, the effect of the electrostatic assistance on the rate of NCL was found to be astonishing. We measured a second order rate constant of $\sim 31000 \text{ M}^{-1} \text{ s}^{-1}$, which can be compared to $\sim 8000 \text{ M}^{-1} \text{ s}^{-1}$ for the diphosphoserine module.

The manuscript was modified accordingly:

Importantly, ligation of the peptide thioester equipped with three phosphoserines, i.e. peptide thioester **1e**, with Cys peptide **7** proceeded at a rate of $31031 \pm 2222 \text{ M}^{-1} \text{ s}^{-1}$. Such a five-digit rate constant, which is significantly higher than the one obtained with only two phosphoserines in the negatively charged module, enabled the ligation to be completed in less than 200 s at 5 μM , with yields as high as 97%. No branched thioester byproduct was observed in this case.

Fig. 4b,c was modified to incorporate the new data.

5. Please delete one “.” in line 6 at page 7.

Corrected

Reviewer #2 (Remarks to the Author):

Ollivier et al. describe the application of electrostatic catalysis to the area of peptide and protein conjugation reactions. The underlying concept of artificially increasing affinity between two reaction partners by positive electrostatic interactions is not new and has seen a certain revival over the past years. Here, the authors take advantage of the bi-dentate interaction of phosphate (on phosphoserine residues) introduced close to the conjugation site on one reaction partner with multiple arginine residues its counterpart. This provides sufficient affinity to bring reactions partners together and to achieve selective reactions between two peptides/proteins at the desired N-terminal amino group of a protein domain and an synthetic peptide carrying a C-terminal thioester in a direct aminolysis reaction. Such reactions have been pioneered by Aimoto et al. for linking synthetic peptides via a native amide bond (this should be referenced in this manuscript) and proceed at very high concentrations only. They also require protection of sidechain amino groups of lysine residues, which is not needed here.

The work of Aimoto on the direct aminolysis of peptide thioesters is now cited in the introduction:

Note that the peptide thioester aminolysis reaction, which involves the reaction of a C-terminal peptide thioester with the α -amino group of another peptide segment, has been successfully used by Hojo and Aimoto for chemical protein synthesis¹². This method uses silver ion in an organic solvent as a mean for activating the thioester and requires the protection of the amine side-chain functionality of lysine residues which, if left unprotected, would compete for the α -amino group.

In addition the authors nicely demonstrate the pseudo-first order reaction kinetics, most likely due to formation of a pre-ligation complex based on electrostatic interactions. Addition of salt such as phosphate of the charged chaotrope Gnd-HCl indicates that in deed electrostatic interactions are facilitating these reactions. The required number of Arg and pSer residues is also analyzed. Furthermore the reaction is extended to native chemical ligation and shows a remarkable increase in reaction kinetics at very low concentrations (1-5 μM), which are typically not within the range of standard NCL reaction (but can be used in DSL and EPSL reaction that should be mentioned as well).

The rates we report in this manuscript hare significantly higher those reported for DSL or other methods relying on acyl donor activation.

We have added this comment:

For comparison, one of the fastest NCL-like process designed so far, the diselenide-selenoester ligation, was shown to reach completion in 24 h at 5 μM ^{29, 30}. In another example, an NCL reaction mediated by an oxalamide thioester surrogate was recently found to be complete in 7 h at the same peptide concentration³¹.

Another example is hydrazine formation even though no data is provided in the main manuscript.

Overall, I really appreciate the very solid study on the use of electrostatic catalysis in peptide and protein ligation and its potential applications. I believe this manuscript could be suited for publication in Nat Comm after the authors have addressed several points:

The title refers to “protein chemical modification”, however the data shown is really on ligation reactions as the electrostatic tags are always placed at the N-or C-termini. Please adjust the title accordingly.

The title was changed a requested:

A biomimetic electrostatic catalysis approach for guiding and promoting N-terminal protein chemical modification

In addition to this, we included the following sentence to highlight the importance of N-terminal protein modifiatiion in conjugate synthesis:

In this work, we selected reactions enabling the N-terminal chemical modification of proteins, a modification site that is often exploited for protein conjugate synthesis^{10, 11, 12, 13}

The conclusions refers to the fact that the Arg-tag can be genetically encoded, opening the way for live cell modifications. However, the authors need to critically discuss that the pSer are not genetically encoded and that they are introduced by chemical synthesis.

We agree. The section “*Design of the charged modules*” was modified as follows (underlined) with appropriate references:

Therefore, we primarily used phosphoserine (pSer) residues to provide negative charges to the reactants, typically in the form of a tripeptidic pSer-Gly-pSer module (Fig. 1B). Because the programming of pSer in live cells is a difficult task,^{12, 13} we also evaluated the interest of glutamic acid residue as the negative charge carrier in some studies. On the other side,...

This also brings up the question why the authors have not tested Asp or Glu to induce the electrostatic interaction (as these are genetically encoded)? Ideally an example with an Asp or Glu tag would be included in this study.

We agree that evaluating Glu as a negative charge carrier broaden the scope of the study.

The pSer/Glu comparison was done using the NCL reaction.

The section “Electrostatic catalysis of the native chemical ligation” was modified as follows:

When phosphoserines were replaced by glutamic acid residues used as negative charge carriers, ligations were successful as well (Fig. 4B,C). The rate achieved by using four glutamate residues (**1g** + **7** → **8g**, $k = 4920 \pm 1300 \text{ M}^{-1} \text{ s}^{-1}$) which bring to the peptide thioester four negative charges as for the pSer-Gly-pSer module is significant. This is an important finding because contrary to pSer, Glu residue is genetically encoded. Extra Glu residues can thus be easily incorporated into proteins using the cell machinery. Nevertheless, the rate achieved with four Glu residues is below that obtained with the two-phosphoserine-based module **1a** (**1a** + **7** → **8a**, $k = 7930 \pm 1400 \text{ M}^{-1} \text{ s}^{-1}$). Although this experiment shows the importance of module overall charge on electrostatic catalysis, it reveals that a phosphoserine residue cannot be fully recapitulated by two glutamic acid residues.

Fig. 4b,c was modified to incorporate the new data.

A more generally important point is the discussion of the pSer and Arg residues needed at the ligation site. Many peptide/protein ligations are setup to avoid such ligation scars, especially such highly charged ones that can easily impact protein folding/function. This needs to be discussed in the manuscript.

Here, options of releasable tags should be considered.

The revision includes the experiments showing that pSer at the ligation junction can be easily dephosphorylated using alkaline phosphatase.

The main manuscript was modified in section "*Proof-of-concept for electrostatic catalysis at the protein level*" as follow:

Because the presence of phosphorylated AAs in the final product is not always desirable, we verified that the treatment of peptide **3a,a** with alkaline phosphatase resulted in its rapid dephosphorylation.

The data can be found in the Supplementary Information.

Now concerning the Arg module, the consequence of having these residues in the final product really depend of the final application. It is difficult to give a general comment on this problem, as having these residues in the product can be an advantages (increased solubility) or a disadvantage (for ex. Immune response directed toward the scar).

What is the authors opinion of their reactions with highly charged proteins? A nice control would be a test reaction with peptides containing additional pSer or Arg residues further away from the reactive termini.

This is an interesting suggestion made also by reviewer 1.

We investigated this question using the NCL reaction and by placing negatively charged residues near the positively charged module. We did that because Cys peptides having such dual modifications are easier to prepare than modified peptide thioesters as originally suggested by reviewer 1. To peptide were prepared to evaluate the impact of introducing pSer and Glu residues.

The following paragraph was inserted in the section "*Electrostatic catalysis of the native chemical ligation*" to discuss this point:

An important question that we also investigated is whether the presence of negatively charged AAs near the positively charged module might abolish the long-range electrostatic attraction mechanism either by competing intramolecularly with the negatively charged module present on the ligation partner, or by repelling it. To investigate this point, we produced two variants of Cys peptide **7** by extending its C-terminus with pSer-Gly-pSer or Glu-Glu-Gly-Glu-Glu sequences. The reaction of these

Cys peptides with peptide thioester **1a** equipped with pSer-Gly-pSer module proceeded similarly according an apparent second order rate constant of $\sim 700 \text{ M}^{-1} \text{ s}^{-1}$ (see Supplementary Methods). We conclude that although the presence of negatively charged AAs near the positively charged module affects the rate of ligation, electrostatic catalysis remains nevertheless significant in this case too.

The Supplementary Information was modified accordingly (preparation of the peptide reactants and results of the ligation experiments).

The acceleration of NCL by including pSer and Arg tags is impressive but formation of the branched product is a concern. Shouldn't electrostatic repulsion of pSer reduce this side product?

This is the only experiment for which we noticed the formation of a branched thioester product. For example, we didn't saw such levels of branched thioester byproduct formation by using the triphosphoserine module or the Glu-based negative modules. With the new data provided in the manuscript regarding the impact of negative charges nearby the positive module, we can understand that the positive module is not fully masked or inhibited by negative charges placed nearby. However, we have no explanation for the more pronounced formation of the branched thioester with the diphosphoserine module.

The data on hydrazone formation from a peptide with C-terminal hydrazide and one with N-terminal aldehyde is hidden in the SI. I believe this needs to be included in the main MS and rates should also be compared to aniline-catalyzed ligation reactions.

This is a good suggestion. A new section was added to the manuscript that describes the data on hydrazone catalysis. The important message, which is taken from our data, is that electrostatic catalysis and classical catalysis of hydrazone formation by amine compounds act synergistically.

We performed a series of experiments to see what happens when arginine or aniline catalysis is combined with electrostatic catalysis. Because aniline strongly absorbs UV light, we monitored the ligations by LC-MS and not by UV when arginine was used as the sole catalyst. The data are presented in Fig. 5.

The importance of hydrazone ligation to the field of bioconjugation and our results can be found in a new section, "*Beyond peptide bond forming reactions* »:

The last question we investigated is whether phosphoserine-arginine electrostatic catalysis can efficiently assist reactions mechanistically unrelated to thioester aminolysis. We showed that this is effectively the case by studying the rate of hydrazone ligation using peptide hydrazides and α -oxo aldehyde peptides as reactants. The ease of accessing protein hydrazides and aldehyde reactants

makes hydrazone ligation a popular conjugation method.²⁴ Indeed, protein hydrazides can be easily accessed by chemical,²⁵ biochemical²⁶ or biological²⁷ means. The α -oxo aldehyde functionality can be easily installed on the N-terminus of proteins as well, typically by oxidation of a seryl residue or by transamination.²⁸ The importance of this chemistry for conjugate synthesis has stimulated the development of a variety of amine catalysts,²⁴ following the pioneering studies on nucleophilic catalysis of Schiff base formation by aniline^{29, 30}. One important aim of nucleophilic catalysis of hydrazone ligation is to achieve fast rates in water at neutral pH.

The hydrazone forming reaction studied in this work is depicted in Fig. 5a. The nucleophile (peptide hydrazide **11**) is associated with the negatively charged module, while the electrophile (peptide α -oxo aldehyde **12**) is with the positively charged module. For the thioester aminolysis and NCL reactions, the nucleophile (Gly or Cys) was associated with the positively charged module. Therefore, compared to the peptide bond forming reactions described in the previous sections, the position of the nucleophile/electrophile relative to the charged modules is inverted. Another important difference is that hydrazone ligation is an equilibrated process, while thioester aminolysis/NCL reactions are irreversible reactions.

The reactions were conducted in phosphate buffer (50 mM) at neutral pH. The results of the kinetic study is presented in Fig. 5b. The kinetic data could be fitted to an apparent second order rate law (dashed lines). Based on the obtained rate constants, hydrazone formation proceeds ~3 times faster with the peptide reactants equipped with the charged modules (**13a,a** vs **13d,g**) when the catalyst is arginine (50 mM)³¹, and 13 times faster with aniline catalyst used at the same concentration. Another series of experiments were performed examined using a larger diversity of charged peptide modules and arginine as catalyst to enable the monitoring of hydrazone formation by UV spectroscopy (see Supplementary Methods). The taken message from this study is that electrostatic catalysis synergize with amine catalysts to promote hydrazone formation.

The original data on hydrazone ligation monitored by UV remain in the SI.

REVIEWERS' COMMENTS

Reviewer #1 (Remarks to the Author):

The authors responded satisfactorily to all of the critical issues raised by the referees. The work can be published in Nat Commun without revision.

Reviewer #2 (Remarks to the Author):

Ollivier et al. have nicely addressed most of the concerns raised during the first round of reviewing. The additional data on charged peptides provides more mechanistic insights and allows to judge the scope of the approach. This is also true for the dephosphorylation by alkaline phosphatase to give serine residues in N-terminal peptides. The revised title also fits the described experiments much better but it is still partially misleading as it contains the word "catalysis". The rate acceleration observed here is based on the electrostatic attraction of both reaction partners but there is no other substance that helps the reaction and remains unchanged, which would be required to have a true catalyst. Therefore, I suggest to change the wording in the title and within the manuscript when catalysis is mentioned.

Upon implementation of these changes, I recommend publication in Nature Communications.

Dear reviewers,

We thank you for the positive comments.

Regarding the term “catalysis” used in our manuscript, we are aware that it does not conform strictly to the usage recommended by IUPAC. We used the term catalysis because it is utilized regularly in the literature to describe rate accelerations provoked by electrostatic interactions. For example, the title of Bender’s paper (ref 6 in the manuscript) is: “Electrostatic Catalysis. The Reactivity of an Ester and a Nucleophile of Opposite Charge”. To give another example linked to the importance of “electrostatic catalysis” in enzyme functioning: “A new paradigm for electrostatic catalysis of radical reactions in vitamin B12 enzymes. <https://www.pnas.org/doi/10.1073/pnas.0702238104> ».

This said, reviewer 2 is fully right and the term catalysis has been replaced by « assistance » everywhere in the manuscript and supplementary information. It has been removed from the title as well.

Best regards
Oleg Melnyk